# TRANSFER LEARNING FOR SEQUENCE TAGGING WITH HIERARCHICAL RECURRENT NETWORKS

**Zhilin Yang, Ruslan Salakhutdinov & William W. Cohen**
School of Computer Science
Carnegie Mellon University
`{zhiliny,rsalakhu,wcohen}@cs.cmu.edu`

## ABSTRACT

Recent papers have shown that neural networks obtain state-of-the-art performance on several different sequence tagging tasks. One appealing property of such systems is their generality, as excellent performance can be achieved with a unified architecture and without task-specific feature engineering. However, it is unclear if such systems can be used for tasks without large amounts of training data. In this paper we explore the problem of transfer learning for neural sequence taggers, where a source task with plentiful annotations (e.g., POS tagging on Penn Treebank) is used to improve performance on a target task with fewer available annotations (e.g., POS tagging for microblogs). We examine the effects of transfer learning for deep hierarchical recurrent networks across domains, applications, and languages, and show that significant improvement can often be obtained. These improvements lead to improvements over the current state-of-the-art on several well-studied tasks.[1]

## 1 INTRODUCTION

Sequence tagging is an important problem in natural language processing, which has wide applications including part-of-speech (POS) tagging, text chunking, and named entity recognition (NER). Given a sequence of words, sequence tagging aims to predict a linguistic tag for each word such as the POS tag.

An important challenge for sequence tagging is how to transfer knowledge from one task to another, which is often referred to as *transfer learning* (Pan & Yang, 2010). Transfer learning can be used in several settings, notably for low-resource languages (Zirikly & Hagiwara, 2015; Wang & Manning, 2014) and low-resource domains such as biomedical corpora (Kim et al., 2003) and Twitter corpora (Ritter et al., 2011)). In these cases, transfer learning can improve performance by taking advantage of more plentiful labels from related tasks. Even on datasets with relatively abundant labels, multi-task transfer can sometimes achieve improvement over state-of-the-art results (Collobert et al., 2011).

Recently, a number of approaches based on deep neural networks have addressed the problem of sequence tagging in an end-to-end manner (Collobert et al., 2011; Lample et al., 2016; Ling et al., 2015; Ma & Hovy, 2016). These neural networks consist of multiple layers of neurons organized in a hierarchy and can transform the input tokens to the output labels without explicit hand-engineered feature extraction. The aforementioned neural networks require minimal assumptions about the task at hand and thus demonstrate significant generality—one single model can be applied to multiple applications in multiple languages without changing the architecture. A natural question is whether the representation learned from one task can be useful for another task. In other words, is there a way we can exploit the generality of neural networks to improve task performance by sharing model parameters and feature representations with another task?

To address the above question, we study the transfer learning setting, which aims to improve the performance on a *target task* by joint training with a *source task*. We present a transfer learning approach based on a deep hierarchical recurrent neural network, which shares the hidden feature repre-

---

[1] Code is available at `https://github.com/kimiyoung/transfer`

sentation and part of the model parameters between the source task and the target task. Our approach combines the objectives of the two tasks and uses gradient-based methods for efficient training. We study cross-domain, cross-application, and cross-lingual transfer, and present a parameter-sharing architecture for each case. Experimental results show that our approach can significantly improve the performance of the target task when the the target task has few labels and is more related to the source task. Furthermore, we show that transfer learning can improve performance over state-of-the-art results even if the amount of labels is relatively abundant.

We have novel contributions in two folds. First, our work is, to the best of our knowledge, the first that focuses on studying the transferability of different layers of representations for hierarchical RNNs. Second, different from previous transfer learning methods that usually focus on one specific transfer setting, our framework exploits different levels of representation sharing and provides a unified framework to handle cross-application, cross-lingual, and cross-domain transfer.

## 2 RELATED WORK

There are two common paradigms for transfer learning for natural language processing (NLP) tasks, *resource-based* transfer and *model-based* transfer. Resource-based transfer utilizes additional linguistic annotations as weak supervision for transfer learning, such as cross-lingual dictionaries (Zirikly & Hagiwara, 2015), corpora (Wang & Manning, 2014), and word alignments (Yarowsky et al., 2001). Resource-based methods demonstrate considerable success in cross-lingual transfer, but are quite sensitive to the scale and quality of the additional resources. Resource-based transfer is mostly limited to cross-lingual transfer in previous works, and there is not extensive research on extending resource-based methods to cross-domain and cross-application settings.

Model-based transfer, on the other hand, does not require additional resources. Model-based transfer exploits the similarity and relatedness between the source task and the target task by adaptively modifying the model architectures, training algorithms, or feature representation. For example, Ando & Zhang (2005) proposed a transfer learning framework that shares structural parameters across multiple tasks, and improve the performance on various tasks including NER; Collobert et al. (2011) presented a task-independent convolutional neural network and employed joint training to transfer knowledge from NER and POS tagging to chunking; Peng & Dredze (2016) studied transfer learning between named entity recognition and word segmentation in Chinese based on recurrent neural networks. Cross-domain transfer, or domain adaptation, is also a well-studied branch of model-based transfer in NLP. Techniques in cross-domain transfer include the design of robust feature representations (Schnabel & Schütze, 2014), co-training (Chen et al., 2011), hierarchical Bayesian prior (Finkel & Manning, 2009), and canonical component analysis (Kim et al., 2015).

While our approach falls into the paradigm of model-based transfer, in contrast to the above methods, our method focuses on exploiting the generality of deep recurrent neural networks and is applicable to transfer between domains, applications, and languages.

Our work builds on previous work on sequence tagging based on deep neural networks. Collobert et al. (2011) develop end-to-end neural networks for sequence tagging without hand-engineered features. Later architectures based on different combinations of convolutional networks and recurrent networks have achieved state-of-the-art results on many tasks (Collobert et al., 2011; Huang et al., 2015; Chiu & Nichols, 2015; Lample et al., 2016; Ma & Hovy, 2016). These models demonstrate significant generality since they can be applied to multiple applications in multiple languages with a unified network architecture and without task-specific feature extraction.

## 3 APPROACH

In this section, we introduce our transfer learning approach. We first introduce an abstract framework for neural sequence tagging, summarizing previous work, and then discuss three different transfer learning architectures.

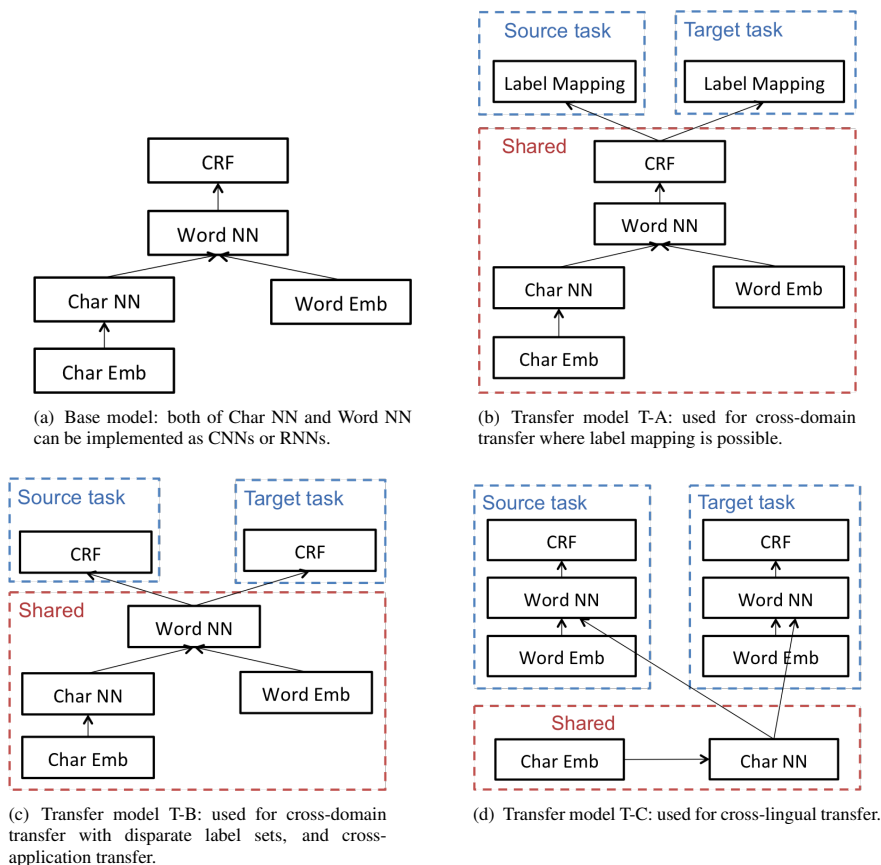

(a) Base model: both of Char NN and Word NN can be implemented as CNNs or RNNs.

(b) Transfer model T-A: used for cross-domain transfer where label mapping is possible.

(c) Transfer model T-B: used for cross-domain transfer with disparate label sets, and cross-application transfer.

(d) Transfer model T-C: used for cross-lingual transfer.

Figure 1: Model architectures: "Char NN" denotes character-level neural networks, "Word NN" denotes word-level neural networks, "Char Emb" and "Word Emb" refer to character embeddings and word embeddings respectively.

## 3.1 BASE MODEL

Though many different variants of neural networks have been proposed for the problem of sequence tagging, we find that most of the models can be described with the hierarchical framework illustrated in Figure 1(a). A character-level layer takes a sequence of characters (represented as embeddings) as input, and outputs a representation that encodes the morphological information at the character level. A word-level layer subsequently combines the character-level feature representation and a word embedding, and further incorporates the contextual information to output a new feature representation. After two levels of feature extraction (encoding), the feature representation output by the word-level layer is fed to a conditional random field (CRF) layer that outputs the label sequence.

Both of the word-level layer and the character-level layer can be implemented as convolutional neural networks (CNNs) or recurrent neural networks (RNNs) (Collobert et al., 2011; Chiu & Nichols, 2015; Lample et al., 2016; Ma & Hovy, 2016). We discuss the details of the model we use in this work in Section 3.4.

## 3.2 TRANSFER LEARNING ARCHITECTURES

We develop three architectures for transfer learning, T-A, T-B, and T-C, are illustrated in Figures 1(b), 1(c), and 1(d) respectively. The three architectures are all extensions of the base model discussed in the previous section with different parameter sharing schemes. We now discuss the use cases for the different architectures.

### 3.2.1 Cross-Domain Transfer

Since different domains are "sub-languages" that have domain-specific regularities, sequence taggers trained on one domain might not have optimal performance on another domain. The goal of cross-domain transfer is to learn a sequence tagger that transfers knowledge from a source domain to a target domain. We assume that few labels are available in the target domain.

There are two cases of cross-domain transfer. The two domains can have label sets that can be mapped to each other, or disparate label sets. For example, POS tags in the Genia biomedical corpus can be mapped to Penn Treebank tags (Barrett & Weber-Jahnke, 2014), while some POS tags in Twitter (e.g., "URL") cannot be mapped to Penn Treebank tags (Ritter et al., 2011).

If the two domains have mappable label sets, we share all the model parameters and feature representation in the neural networks, including the word and character embedding, the word-level layer, the character-level layer, and the CRF layer. We perform a label mapping step on top of the CRF layer. This becomes the model T-A as shown in Figure 1(b).

If the two domains have disparate label sets, we untie the parameter sharing in the CRF layer—i.e., each task learns a separate CRF layer. This parameter sharing scheme reduces to model T-B as shown in Figure 1(c).

### 3.2.2 Cross-Application Transfer

Sequence tagging has a couple of applications including POS tagging, chunking, and named entity recognition. Similar to the motivation in (Collobert et al., 2011), it is usually desirable to exploit the underlying similarities and regularities of different applications, and improve the performance of one application via joint training with another. Moreover, transfer between multiple applications can be helpful when the labels are limited.

In the cross-application setting, we assume that multiple applications are in the same language. Since different applications share the same alphabet, the case is similar to cross-domain transfer with disparate label sets. We adopt the architecture of model T-B for cross-application transfer learning where only the CRF layers are disjoint for different applications.

### 3.2.3 Cross-Lingual Transfer

Though cross-lingual transfer is usually accomplished with additional multi-lingual resources, these methods are sensitive to the size and quality of the additional resources (Yarowsky et al., 2001; Wang & Manning, 2014). In this work, instead, we explore a complementary method that exploits the cross-lingual regularities purely on the model level.

Our approach focuses on transfer learning between languages with similar alphabets, such as English and Spanish, since it is very difficult for transfer learning between languages with disparate alphabets (e.g., English and Chinese) to work without additional resources (Zirikly & Hagiwara, 2015).

Model-level transfer learning is achieved through exploiting the morphologies shared by the two languages. For example, "Canada" in English and "Canadá" in Spanish refer to the same named entity, and the morphological similarities can be leveraged for NER and also POS tagging with nouns. Thus we share the character embeddings and the character-level layer between different languages for transfer learning, which is illustrated as the model T-C in Figure 1(d).

### 3.3 Training

In the above sections, we introduced three neural architectures with different parameter sharing schemes, designed for different transfer learning settings. Now we describe how we train the neural networks jointly for two tasks.

Suppose we are transferring from a source task $s$ to a target task $t$, with the training instances being $X_s$ and $X_t$. Let $W_s$ and $W_t$ denote the set of model parameters for the source and target tasks respectively. The model parameters are divided into two sets, *task specific parameters* and *shared parameters*, i.e.,

$$W_s = W_{s,\text{spec}} \cup W_{\text{shared}}, W_t = W_{t,\text{spec}} \cup W_{\text{shared}},$$

where shared parameters $W_{\text{shared}}$ are jointly optimized by the two tasks, while task specific parameters $W_{s,\text{spec}}$ and $W_{t,\text{spec}}$ are trained for each task separately.

The training procedure is as follows. At each iteration, we sample a task (i.e., either $s$ or $t$) from $\{s, t\}$ based on a binomial distribution (the binomial probability is set as a hyperparameter). Given the sampled task, we sample a batch of training instances from the given task, and then perform a gradient update according to the loss function of the given task. We update both the shared parameters and the task specific parameters. We repeat the above iterations until stopping. We adopt AdaGrad (Duchi et al., 2011) to dynamically compute the learning rates for each iteration. Since the source and target tasks might have different convergence rates, we do early stopping on the target task performance.

## 3.4 Model Implementation

In this section, we describe our implementation of the base model. Both the character-level and word-level neural networks are implemented as RNNs. More specifically, we employ gated recurrent units (GRUs) (Cho et al., 2014). Let $(\mathbf{x}_1, \mathbf{x}_2, \cdots, \mathbf{x}_T)$ be a sequence of inputs that can be embeddings or hidden states of other layers. Let $\mathbf{h}_t$ be the GRU hidden state at time step $t$. Formally, a GRU unit at time step $t$ can be expressed as

$$
\begin{aligned}
\mathbf{r}_t &= \sigma(W_{rx}\mathbf{x}_t + W_{rh}\mathbf{h}_{t-1}) \\
\mathbf{z}_t &= \sigma(W_{zx}\mathbf{x}_t + W_{zh}\mathbf{h}_{t-1}) \\
\tilde{\mathbf{h}}_t &= \tanh(W_{hx}\mathbf{x}_t + W_{hh}(\mathbf{r}_t \odot \mathbf{h}_{t-1})) \\
\mathbf{h}_t &= \mathbf{z}_t \odot \mathbf{h}_{t-1} + (1 - \mathbf{z}_t) \odot \tilde{\mathbf{h}}_t,
\end{aligned}
$$

where $W$'s are model parameters of each unit, $\tilde{\mathbf{h}}_t$ is a candidate hidden state that is used to compute $\mathbf{h}_t$, $\sigma$ is an element-wise sigmoid logistic function defined as $\sigma(\mathbf{x}) = 1/(1 + e^{-\mathbf{x}})$, and $\odot$ denotes element-wise multiplication of two vectors. Intuitively, the update gate $\mathbf{z}_t$ controls how much the unit updates its hidden state, and the reset gate $\mathbf{r}_t$ determines how much information from the previous hidden state needs to be reset. The input to the character-level GRUs is character embeddings, while the input to the word-level GRUs is the concatenation of character-level GRU hidden states and word embeddings. Both GRUs are bi-directional and have two layers.

Given an input sequence of words, the word-level GRUs and the character-level GRUs together learn a feature representation $\mathbf{h}_t$ for the $t$-th word in the sequence, which forms a sequence $\mathbf{h} = (\mathbf{h}_1, \mathbf{h}_2, \cdots, \mathbf{h}_T)$. Let $y = (y_1, y_2, \cdots, y_T)$ denote the tag sequence. Given the feature representation $\mathbf{h}$ and the tag sequence $\mathbf{y}$ for each training instance, the CRF layer defines the objective function to maximize based on a max-margin principle (Gimpel & Smith, 2010) as:

$$
f(\mathbf{h}, \mathbf{y}) - \log \sum_{\mathbf{y}' \in \mathcal{Y}(\mathbf{h})} \exp(f(\mathbf{h}, \mathbf{y}') + \text{cost}(\mathbf{y}, \mathbf{y}')),
$$

where $f$ is a function that assigns a score for each pair of $\mathbf{h}$ and $\mathbf{y}$, and $\mathcal{Y}(\mathbf{h})$ denotes the space of tag sequences for $\mathbf{h}$. The cost function $\text{cost}(\mathbf{y}, \mathbf{y}')$ is added based on the max-margin principle (Gimpel & Smith, 2010) that high-cost tags $\mathbf{y}'$ should be penalized more heavily.

Our base model is similar to Lample et al. (2016), but in contrast to their model, we employ GRUs for the character-level and word-level networks instead of Long Short-Term Memory (LSTM) units, and define the objective function based on the max-margin principle. We note that our transfer learning framework does not make assumptions about specific model implementation, and could be applied to other neural architectures (Collobert et al., 2011; Chiu & Nichols, 2015; Lample et al., 2016; Ma & Hovy, 2016) as well.

## 4 Experiments

### 4.1 Datasets

We use the following benchmark datasets in our experiments: Penn Treebank (PTB) POS tagging, CoNLL 2000 chunking, CoNLL 2003 English NER, CoNLL 2002 Dutch NER, CoNLL 2002 Spanish NER, the Genia biomedical corpus (Kim et al., 2003), and a Twitter corpus (Ritter et al., 2011).

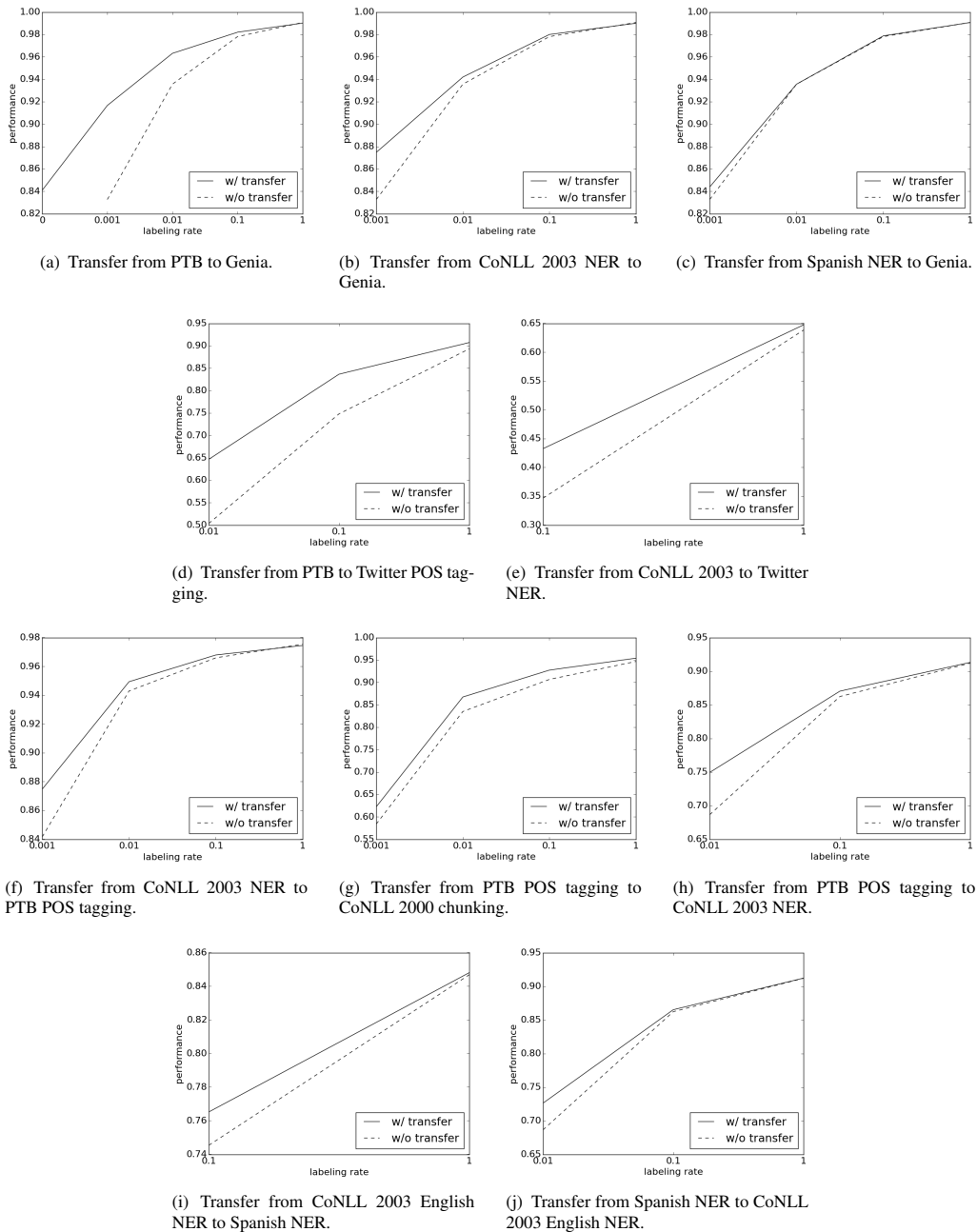

(a) Transfer from PTB to Genia.

(b) Transfer from CoNLL 2003 NER to Genia.

(c) Transfer from Spanish NER to Genia.

(d) Transfer from PTB to Twitter POS tagging.

(e) Transfer from CoNLL 2003 to Twitter NER.

(f) Transfer from CoNLL 2003 NER to PTB POS tagging.

(g) Transfer from PTB POS tagging to CoNLL 2000 chunking.

(h) Transfer from PTB POS tagging to CoNLL 2003 NER.

(i) Transfer from CoNLL 2003 English NER to Spanish NER.

(j) Transfer from Spanish NER to CoNLL 2003 English NER.

Figure 2: Results on transfer learning. Cross-domain transfer: Figures 2(a), 2(d), and 2(e). Cross-application transfer: Figures 2(f), 2(g), and 2(h). Cross-lingual transfer: Figures 2(i) and 2(j). Transfer across domains and applications: Figure 2(b). Transfer across domains, applications, and languages: Figure 2(c).

The statistics of the datasets are described in Table 1. We construct the POS tagging dataset with the instructions described in Toutanova et al. (2003). Note that as a standard practice, the POS tags are extracted from the parsed trees. For the CoNLL 2003 English NER dataset, we follow previous works (Collobert et al., 2011) to append one-hot gazetteer features to the input of the CRF layer for fair comparison. Since there is no standard training/dev/test data split for the Genia and Twitter corpora, we randomly sample 10% for test, 10% for development, and 80% for training. We follow previous work (Barrett & Weber-Jahnke, 2014) to map Genia POS tags to PTB POS tags.

Table 1: Dataset statistics.

| Benchmark | Task | Language | # Training Tokens | # Dev Tokens | # Test Tokens |
|---|---|---|---|---|---|
| PTB 2003 | POS Tagging | English | 912,344 | 131,768 | 129,654 |
| CoNLL 2000 | Chunking | English | 211,727 | - | 47,377 |
| CoNLL 2003 | NER | English | 204,567 | 51,578 | 46,666 |
| CoNLL 2002 | NER | Dutch | 202,931 | 37,761 | 68,994 |
| CoNLL 2002 | NER | Spanish | 207,484 | 51,645 | 52,098 |
| Genia | POS Tagging | English | 400,658 | 50,525 | 49,761 |
| Twitter | POS Tagging | English | 12,196 | 1,362 | 1,627 |
| Twitter | NER | English | 36,936 | 4,612 | 4,921 |

Table 2: Improvements with transfer learning under multiple low-resource settings (%). "Dom", "app", and "ling" denote cross-domain, cross-application, and cross-lingual transfer settings respectively. The numbers following the slashes are labeling rates (chosen such that the number of labeled examples are of the same scale).

| Source | Target | Model | Setting | Transfer | No Transfer | Delta |
|---|---|---|---|---|---|---|
| PTB | Twitter/0.1 | T-A | dom | 83.65 | 74.80 | 8.85 |
| CoNLL03 | Twitter/0.1 | T-A | dom | 43.24 | 34.65 | 8.59 |
| PTB | CoNLL03/0.01 | T-B | app | 74.92 | 68.64 | 6.28 |
| PTB | CoNLL00/0.01 | T-B | app | 86.73 | 83.49 | 3.24 |
| CoNLL03 | PTB/0.001 | T-B | app | 87.47 | 84.16 | 3.31 |
| Spanish | CoNLL03/0.01 | T-C | ling | 72.61 | 68.64 | 3.97 |
| CoNLL03 | Spanish/0.01 | T-C | ling | 60.43 | 59.84 | 0.59 |
| PTB | Genia/0.001 | T-A | dom | 92.62 | 83.26 | 9.36 |
| CoNLL03 | Genia/0.001 | T-B | dom&app | 87.47 | 83.26 | 4.21 |
| Spanish | Genia/0.001 | T-C | dom&app&ling | 84.39 | 83.26 | 1.13 |
| PTB | Genia/0.001 | T-B | dom | 89.77 | 83.26 | 6.51 |
| PTB | Genia/0.001 | T-C | dom | 84.65 | 83.26 | 1.39 |

## 4.2 TRANSFER LEARNING PERFORMANCE

We evaluate our transfer learning approach on the above datasets. We fix the hyperparameters for all the results reported in this section: we set the character embedding dimension at $25$, the word embedding dimension at $50$ for English and $64$ for Spanish, the dimension of hidden states of the character-level GRUs at $80$, the dimension of hidden states of the word-level GRUs at $300$, and the initial learning rate at $0.01$. Except for the Twitter datasets, these datasets are fairly large. To simulate a low-resource setting, we also use random subsets of the data. We vary the labeling rate of the target task at $0.001$, $0.01$, $0.1$ and $1.0$. Given a labeling rate $r$, we randomly sample a ratio $r$ of the sentences from the training set and discard the rest of the training data—e.g., a labeling rate of $0.001$ results in around 900 training tokens on PTB POS tagging (Cf. Table 1).

The results on transfer learning are plotted in Figure 2, where we compare the results with and without transfer learning under various labeling rates. The numbers in the y-axes are accuracies for POS tagging, and chunk-level F1 scores for chunking and NER. The numbers are shown in Table 2. We can see that our transfer learning approach consistently improved over the non-transfer results. We also observe that the improvement by transfer learning is more substantial when the labeling rate is lower. For cross-domain transfer, we obtained substantial improvement on the Genia and Twitter corpora by transferring the knowledge from PTB POS tagging and CoNLL 2003 NER. For example, as shown in Figure 2(a), we can obtain an tagging accuracy of $83\%+$ with zero labels and $92\%$ with only $0.001$ labels when transferring from PTB to Genia. As shown in Figures 2(d) and 2(e), our transfer learning approach can improve the performance on Twitter POS tagging and NER for all labeling rates, and the improvements with $0.1$ labels are more than $8\%$ for both datasets. Cross-application transfer also leads to substantial improvement under low-resource conditions. For

Table 3: Comparison with state-of-the-art results (%).

| Model | CoNLL 2000 | CoNLL 2003 | Spanish | Dutch | PTB 2003 |
|---|---|---|---|---|---|
| Collobert et al. (2011) | 94.32 | 89.59 | – | – | 97.29 |
| Passos et al. (2014) | – | 90.90 | – | – | – |
| Luo et al. (2015) | – | 91.2 | – | – | – |
| Huang et al. (2015) | 94.46 | 90.10 | – | – | 97.55 |
| Gillick et al. (2015) | – | 86.50 | 82.95 | 82.84 | – |
| Ling et al. (2015) | – | – | – | – | **97.78** |
| Lample et al. (2016) | – | 90.94 | 85.75 | 81.74 | – |
| Ma & Hovy (2016) | – | 91.21 | – | – | 97.55 |
| Ours w/o transfer | 94.66 | 91.20 | 84.69 | 85.00 | 97.55 |
| Ours w/ transfer | **95.41** | **91.26** | **85.77** | **85.19** | 97.55 |

example, as shown in Figures 2(g) and 2(h), the improvements with 0.1 labels are 6% and 3% on CoNLL 2000 chunking and CoNLL 2003 NER respectively when transferring from PTB POS tagging. Figures 2(j) and 2(i) show that cross-lingual transfer can improve the performance when few labels are available.

Figure 2 further shows that the improvements by different architectures are in the following order: T-A > T-B > T-C. This phenomenon can be explained by the fact that T-A shares the most model parameters while T-C shares the least. Transfer settings like cross-lingual transfer can only use T-C because the underlying similarities between the source task and the target task are less prominent (i.e., less *transferable*), and in those cases the improvement by transfer learning is less substantial.

Another interesting comparison is among Figures 2(a), 2(b), and 2(c). Figure 2(a) is cross-domain transfer, Figure 2(b) is transfer across domains and applications at the same time, and Figure 2(c) combines all the three transfer settings (i.e., from Spanish NER in the general domain to English POS tagging in the biomedical domain). The results show that the improvement by transfer learning diminishes when the transfer becomes "indirect" (i.e., the source task and the target task are more loosely related).

We also study using different transfer learning models for the same task. We study the effects of using T-A, T-B, and T-C when transferring from PTB to Genia, and the results are included in the lower part of Table 2. We observe that the performance gain decreases when less parameters are shared (i.e., T-A > T-B > T-C).

## 4.3 COMPARISON WITH STATE-OF-THE-ART RESULTS

In the above section, we examine the effects of different transfer learning architectures. Now we compare our approach with state-of-the-art systems on these datasets.

We use publicly available pretrained word embeddings as initialization. On the English datasets, following previous works that are based on neural networks (Collobert et al., 2011; Huang et al., 2015; Chiu & Nichols, 2015; Ma & Hovy, 2016), we experiment with both the 50-dimensional SENNA embeddings (Collobert et al., 2011) and the 100-dimensional GloVe embeddings (Pennington et al., 2014) and use the development set to choose the embeddings for different tasks and settings. For Spanish and Dutch, we use the 64-dimensional Polyglot embeddings (Al-Rfou et al., 2013). We set the hidden state dimensions to be 300 for the word-level GRU. The initial learning rate for AdaGrad is fixed at 0.01. We use the development set to tune the other hyperparameters of our model.

Our results are reported in Table 3. Since there are no standard data splits on the Genia and Twitter corpora, we do not include these datasets into our comparison. The results for CoNLL 2000 chunking, CoNLL 2003 NER, and PTB POS tagging are obtained by transfer learning between the three tasks, i.e., transferring from two tasks to the other. The results for Spanish and Dutch NER are obtained with transfer learning between the NER datasets in three languages (English, Spanish, and Dutch). From Table 3, we can draw two conclusions. First, our transfer learning approach achieves new state-of-the-art results on all the considered benchmark datasets except PTB POS tagging, which indicates that transfer learning can still improve the performance even on datasets with

relatively abundant labels. Second, our base model (w/o transfer) performs competitively compared to the state-of-the-art systems, which means that the improvements shown in Section 4.2 are obtained over a strong baseline.

## 5   CONCLUSION

In this paper we develop a transfer learning approach for sequence tagging, which exploits the generality demonstrated by deep neural networks in previous work. We design three neural network architectures for the settings of cross-domain, cross-application, and cross-lingual transfer. Our transfer learning approach achieves significant improvement on various datasets under low-resource conditions, as well as new state-of-the-art results on some of the benchmarks. With thorough experiments, we observe that the following factors are crucial for the performance of our transfer learning approach: a) label abundance for the target task, b) relatedness between the source and target tasks, and c) the number of parameters that can be shared. In the future, it will be interesting to combine model-based transfer (as in this work) with resource-based transfer for cross-lingual transfer learning.

ACKNOWLEDGMENTS

This work was funded by NVIDIA, the Office of Naval Research grant N000141512791, the ADeLAIDE grant FA8750-16C-0130-001, the NSF grant IIS1250956, and Google Research.

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
