# Peer review of "Transfer Learning for Sequence Tagging with Hierarchical Recurrent Networks"

_ICLR 2017 — accepted_

[Official Review · AnonReviewer1 · rating 8 · confidence 4 · 15 Dec 2016]
**Accept**

This paper presents a clear hierarchical taxonomy of transfer learning methods as applicable to sequence tagging problems. This contextualizes and unifies previous work on specific instances of this taxonomy. Moreover, the paper shows that previously unexplored places in this taxonomy are competitive with or superior to the state of the art in key benchmark problems.

It'd be nice to see this explored further, such as highlighting what is the loss as you move from the more restrictive to the less restrictive transfer learning approaches, but I believe this paper is interesting and acceptable as-is.

[Official Review · AnonReviewer4 · rating 7 · confidence 4 · 16 Dec 2016 (modified: 20 Jan 2017)]
**Accept**

Authors' response well answered my questions. Thanks!
Evaluation not changed.

###

This paper proposes a hierarchical framework of transfer learning for sequence tagging, which is expected to help the target task with the source task, by sharing as many levels of representation as possible. It is a general framework for various neural models. The paper has extensive and solid experiments, and the performance is competitive with the state of the art on multiple benchmark datasets. The framework is clear by itself, except that more details about training procedure, i.e. sec-3.3, need to be added. 

The experimental results show that for some task pairs {s,t}, this framework can help low-resource target task t, and the improvement increases with more levels of representations can be shared. Firstly, I suggest that the terms *source* and *target* should be more precisely defined in the current framework, because, due to Sec-3.3, the s and t in each pair are sort of interchangeable. That is, either of them can be the *source* or *target* task, especially when p(X=s)=p(X=t)=0.5 is used in the task sampling. The difference is: one is low-resourced and the other is not. Thus it could be thought of as multi-tasking between tasks with imbalanced resource. So one question is: does this framework simultaneously help both tasks in the pair, by learning more generalizable representations for different domains/applications/languages? Or is it mostly likely to only help the low-resourced one? Does it come with sacrifice on the high-resourced side? 

Secondly, as the paper shows that the low-resourced tasks are improved for the selected task pairs, it would also be interesting and helpful to know how often this could happen. That is, when the tasks are randomly paired (one chosen from a low-resource pool and the other from a high resource pool), how often could this framework help the low-resourced one?

Moreover, the choice of T-A/T-B/T-C lies intuitively in how many levels of representation *could* be shared as possible. This implicitly assumes share more, help more. Although I tend to believe so, it would be interesting to have some empirical comparison. For example, one could perhaps select some cross-domain pair, and see if T-A > T-B > T-C on such pairs, as mentioned in the author’s answer to the pre-review question. 

In general, I think this is a solid paper, and more exploration could be done in this direction. So I tend to accept this paper.

[Author Response · Zhilin Yang · 18 Dec 2016]
**Table 2 updated to include comparison of different architectures for one task pair.**

We study the effects of using T-A, T-B, and T-C when transferring from PTB to Genia. The results are included in the lower part of Table 2. It is clear that the performance gain decreases when less parameters are shared.

[Official Review · AnonReviewer3 · rating 5 · confidence 4 · 18 Dec 2016]
**Interesting research direction, but the scientific advances are limited and the experiments are not very convincing.**
substance 2 · meaningful comparison 4

The authors propose transfer learning variants for neural-net-based models, applied to a bunch of NLP tagging tasks.

The field of multi-tasking is huge, and the approaches proposed here do not seem to be very novel in terms of machine learning: parts of a general architecture for NLP are shared, the amount of shared "layers" being dependent of the task of interest.

The novelty lies in the type of architecture which is used in the particular setup of NLP tagging tasks.

The experimental results show that the approach seems to work well when there is not much labeled data available (Figure 2). Table 3 show some limited improvement at full scale.

Figure 2 results are debatable though: it seems the authors fixed the architecture size while varying the amount of labeled data; it is very likely that tuning the architecture for each size would have led to better results.

Overall, while the paper reads well, the novelty seems a bit limited and the experimental section seems a bit disappointing.

[Final Decision · Program Chairs · 06 Feb 2017]
**ICLR committee final decision**

One weak and one positive review without much concrete substance. The third review is positive, but the experiments are not that convincing: the gains from transfer are small in table 3 and in table 2 it is unclear how strong the baselines are. Given how competitive ICLR is, the area chair has no alternative than to unfortunately reject this paper.